# Satellite altimetry and operational oceanography: from Jason-1 to SWOT

Pierre-Yves Le Traon[1,2], Gérald Dibarboure[3], Jean-Michel Lellouche[1], Marie-Isabelle Pujol[4], Mounir Benkiran[1], Marie Drevillon[1], Yann Drillet[1], Yannice Faugère[3], Elisabeth Remy[1]

[1]Mercator Ocean International, Toulouse, 31400, France
[2]Ifremer, Plouzané, 29280, France
[3]Centre National d'Études Spatiales, Toulouse, 31400, France
[4]CLS, Ramonville St Agne, 31520, France

*Correspondence to*: P.Y. Le Traon (pierre-yves.letraon@mercator-ocean.fr)

**Abstract.** The development and evolution of satellite altimetry and operational oceanography are very closely linked. By providing all weather, global and real time observations of sea level, a key variable to constrain ocean analysis and forecasting systems, satellite altimetry has had a profound influence on the advancement of operational oceanography. Over the past 20 years, satellite altimetry has been providing a continuous observation of the ocean in near real time. From the launch of Jason-1 in 2001 to the launch of SWOT in 2022, satellite altimetry capabilities have been regularly improved through refinements of geophysical corrections, processing algorithms including real time processing, and evolution of altimeter radar technology (SAR mode, swath altimetry). Resolution has also improved through the use of multiple altimeters and now swath altimetry. In parallel, major improvements of ocean prediction systems have occurred from the GODAE demonstration in the 2000s up to fully operational systems now serving a large range of applications in the 2020s. The paper provides an overview of the development and evolution of satellite altimetry and operational oceanography over the past 20 years in the context of the Mercator Ocean prediction center, the DUACS system, and the EU Copernicus Marine Service. Impact of altimetry on the performances of ocean prediction systems (based on OSEs and OSSEs) is reviewed. The future contribution of swath altimetry on ocean prediction is also discussed. Prospects for the next decade are addressed in the conclusion.

## 1 Introduction

The launch of TOPEX/Poseidon (T/P) in 1992 was a breakthrough in the development of satellite altimetry. Thanks to a fully optimized altimeter mission, the large-scale sea level and ocean circulation variations were observed from the first time from space. This led to many discoveries (see Fu and Cazenave, 2001). The ERS-1/2 missions flew together with T/P and provided complementary sampling required to monitor the mesoscale circulation. The harmonization, intercalibration and combination of the T/P with ERS-1/2 (Ducet et al., 2000) was a major step to develop the use of satellite altimetry in models (see Le Traon, 2013). The launch of Jason-1 in 2001 was then a major milestone to start a series of highly precise long-term

reference missions (following T/P) with near real time processing capabilities. Since then, the ocean has been continuously monitored with multiple altimeter missions (Fig. 1).

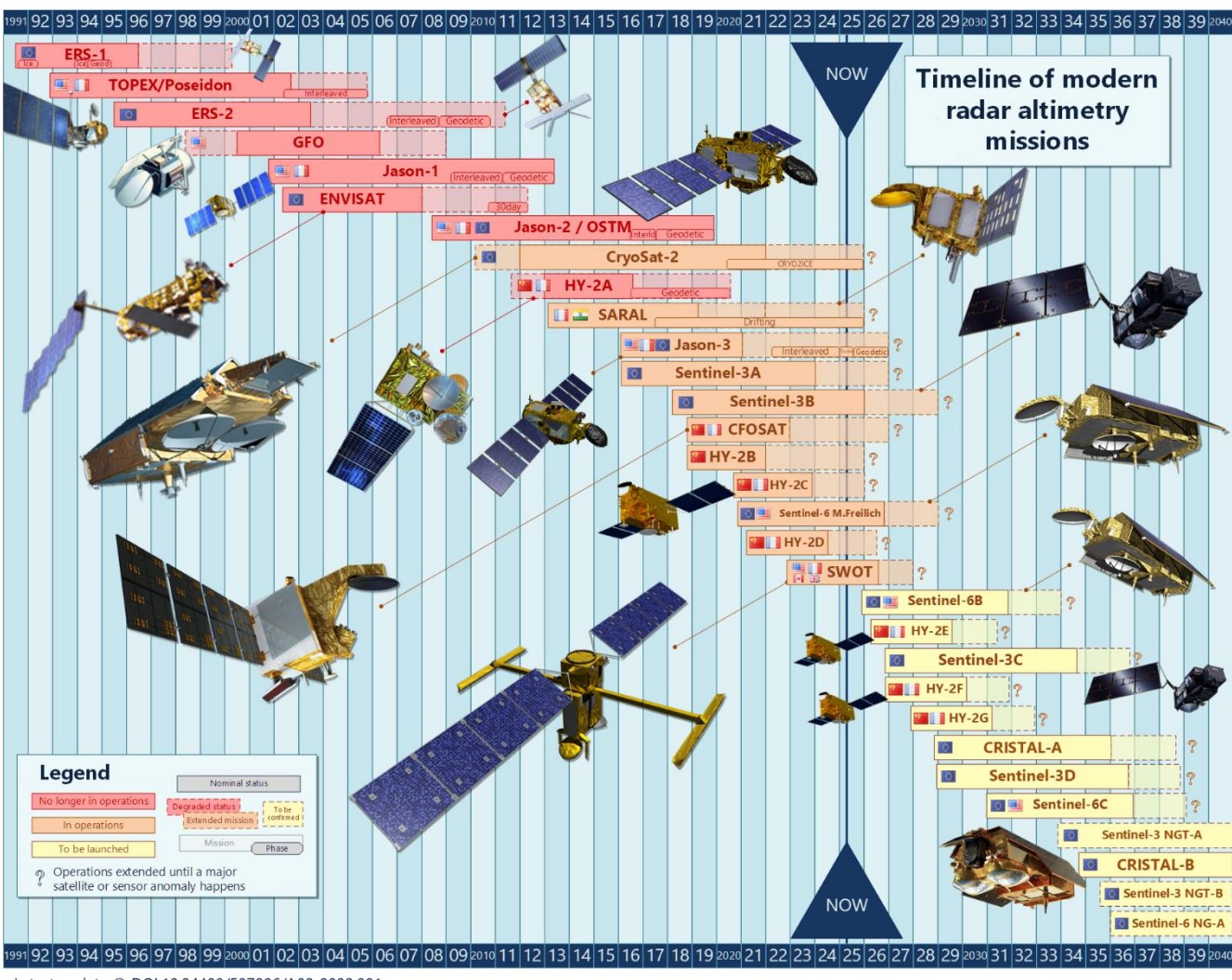

Latest update @ DOI:10.24400/527896/A02-2022.001

**Figure 1: Timeline of modern radar altimetry missions. Adapted from doi://10.24400/527896/A02-2022.001 version 2024/03.**

The development of global operational oceanography started with the Global Data Assimilation Experiment (GODAE) (Smith and Lefebvre, 1997) and its operational demonstration in the early 2000s (Le Traon et al., 2001). The demonstration was phased in with the Jason-1 and ENVISAT altimeter missions that provided the required real-time observation of the global ocean. The start of Argo, the global array of profiling floats, was required to provide the essential complementary observation of the ocean interior (Roemmich et al., 2001). Following GODAE, operational oceanography systems have been developed from global to regional and coastal scales (e.g. Bell et al., 2015; Metzger et al., 2014; Chassignet et al., 2018;

Hirose et al., 2019). There have been major advancements in operational oceanography systems. The resolution of global ocean analysis and forecasting systems has increased by a factor of at least 3 from 1/4° to 1/12° or more now (e.g. Lellouche et al., 2018). Regional systems have reached a resolution of a few kilometers (e.g. Francis et al., 2020; Ciliberti et al., 2022). Models now use high-frequency atmospheric forcing (< 3 h), tidal forcing and improved ocean/wave/atmosphere coupling. More advanced data assimilation techniques have been developed with better characterization of model error covariances including the use of ensemble approaches. All these improvements have led to better use of altimeter data in models.

The development of the altimeter constellation has been an important driver for the evolution of the operational oceanography systems. Satellite altimetry plays a prominent role in constraining ocean models through data assimilation (e.g. Le Traon et al., 2017). It provides global, real time, all-weather sea level measurements with high space and time resolution. Sea level is an integral of the ocean interior properties (sea level is related to depth-integrated density perturbations and barotropic motions) and is a strong constraint for inferring the 4D ocean circulation through data assimilation. Only altimetry can constrain the 4D mesoscale circulation in ocean models which is required for most operational oceanography applications. The use of multiple altimeter missions has been a common feature of all operational oceanography systems (see Le Traon et al., 2017). Major progress was made to ensure homogenization, intercalibration and real time / delayed mode processing of altimeter missions before data are assimilated in models either in forecast or reanalysis modes (e.g. Dibarboure et al., 2011; Pujol et al., 2016). Assimilation of altimeter data in models requires precise Mean Dynamic Topographies (MDTs) to reference the altimeter Sea Level Anomaly (SLA) observations. MDTs have been greatly improved thanks to the use of GOCE and GRACE geoid data and in-situ observations (e.g. Rio et al., 2014; Mulet et al., 2021).

The paper reviews the joint development of satellite altimetry and operational oceanography over the past 20 years. It is organized as follows. Section 2 provides an overview of the development and evolution of satellite altimetry and operational oceanography from the DUACS, Mercator Ocean and Copernicus Marine perspective. Impact of altimetry on the performances of ocean prediction systems (based on OSEs and OSSEs) is reviewed in section 3. Section 4 discusses the future contribution of swath altimetry for ocean prediction. Prospects for the next decade are presented in section 5.

**2 Development and evolution of satellite altimetry and operational oceanography**

An overview of the development of the altimeter constellation, altimeter data processing systems and operational oceanography is given in the next sections. This overview extends previous syntheses given in Dibarboure et al. (2011) and Le Traon et al. (2017). It focuses on capabilities developed in the framework of the EU Copernicus Marine Service, the DUACS system, and the Mercator Ocean prediction center.

## 2.1 Altimeter constellation

The minimum requirement for operational oceanography is 3 to 4 missions, and sometimes more if one takes into account the inevitable data outage and anomalies happening during the lifetime of all satellites (Le Traon et al., 2017). Dibarboure and Lambin (2015) explain that constellation efficiency is not just related to the number of altimeters. Indeed, one needs to consider the time needed before an altimeter becomes operational (also known a commissioning phase or Calibration/Validation phase) as well as satellite sampling redundancy (Dibarboure and Morrow, 2016): two altimeters operated on similar orbits might provide duplicate measurements (e.g. tandem phase of Jason-3 and Sentinel-6MF). Moreover, some altimeter missions are not optimized for operational oceanography because they use geodetic orbits, have non optimal payloads for orbit determination or lack a microwave radiometer or dual frequency system, for example. Some satellites "of opportunity" can provide sea surface height but with a larger error budget or degraded coverage with respect to operational missions.

In that context, there have been considerable changes in the radar altimeter constellation over the past 15 years, both in terms of number of satellites and technology. From 2010 to 2015, the altimeter constellation was in a very fragile state, and operational oceanography relied primarily on non-operational altimeters. In 2012, altimeters from the first couple of modern generations had been decommissioned, and only Jason-2 remained. Thanks to the coordinated efforts of space agencies and operational oceanography, the critical number of 3-4 satellites (Le Traon et al., 2017) was secured with CRYOSAT-2 (originally a cryosphere-oriented altimeter mission) and Haiyang-2A (originally China's first technology demonstrator) and then SARAL (originally the first Ka-band altimeter technology demonstrator from France & India). Despite periodic data outages (Fig. 2), the constellation proved sufficiently robust to maintain decent observational quality for the operational models.

a)

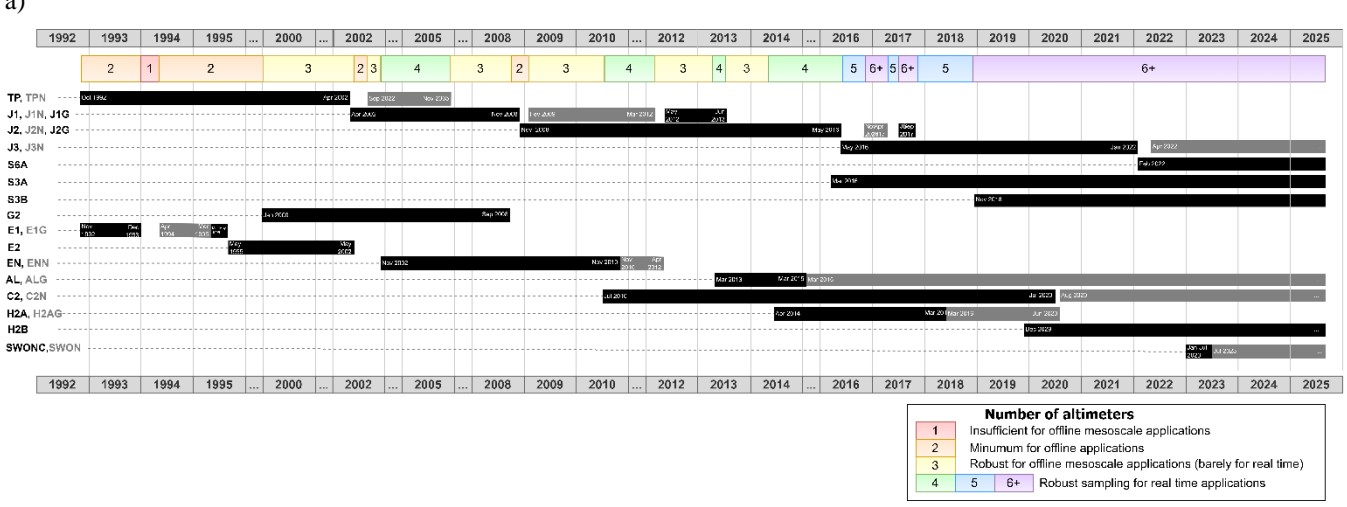

b)

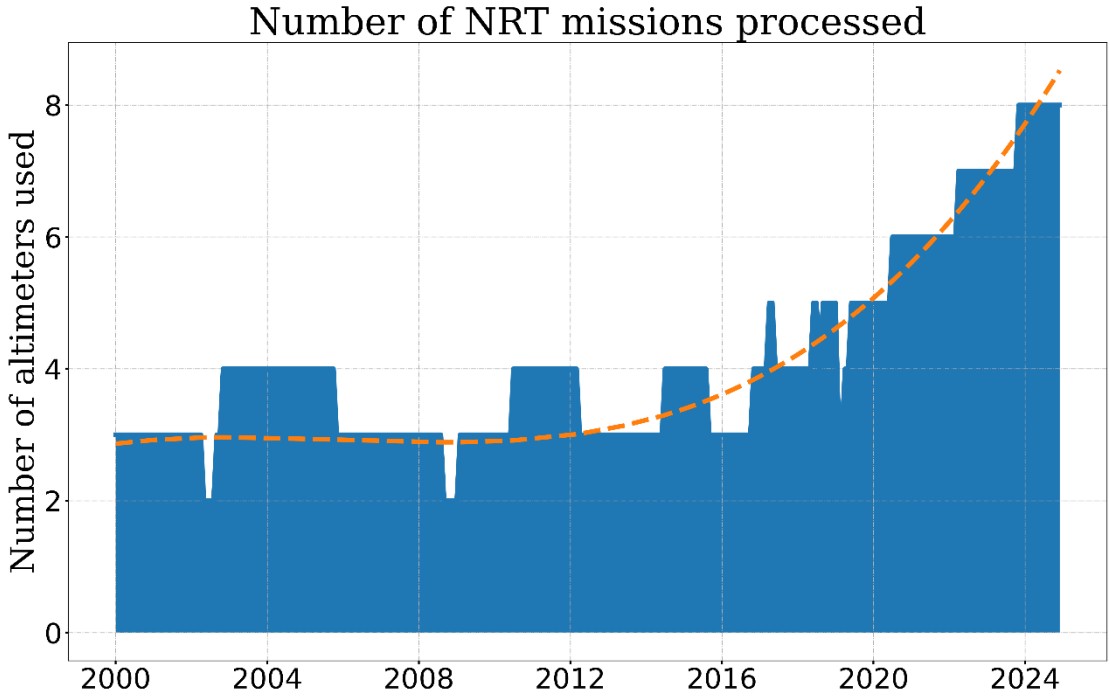

**Figure 2: Evolution of the number of altimeter missions injected in Copernicus Sea Level Thematic Assembly Center (also known as DUACS). Panel (a) is for the offline/reprocessed products, and panel (b) is the monthly availability used for the near real time product. Adapted from the quality information document CMEMS-SL-QUID-008-032-068 (doi://10.48670/moi-00149).**

From 2014 onwards, the following generation of operational missions progressively strengthened the constellation. By the late 2010s, Jason-3 and Sentinel-3A provided the critical core observation in addition to the aforementioned satellites "of opportunity", and new additions replaced ageing satellites (e.g. Haiyang-2B replaced its predecessor). About 6 or 7 satellites were always active (Figure 2a), although one or two were kept as backup in operational systems (on standby, qualified for

service, with known limitations or complexity). In the early 2020s, the numbers went up again as older satellites were still working nominally while the following generation joined the constellation (Haiyang-2B and 2C, Jason-CS/Sentinel-6 Michael Freilich, and SWOT). At the time of this writing, there are 11 altimeters in operations: one third are from the operational Copernicus programme in Europe, one third are from the Haiyang programme from China, and the others are technology demonstrators.

In the coming years, the constellation should remain very solid as new satellites are about to be launched to replace older missions (Sentinel-3C, Sentinel-6B, Haiyang-2E) until the next generation of mission takes over (CRISTAL, Sentinel-3NG, Sentinel-6NG) within the next 10 years. In that context, ocean prediction systems can expect to obtain not only the minimum coverage needed to carry out their mission, but also denser observational coverage able to resolve scales finer in space or in time (Ballarotta et al. 2019; Pujol et al. 2023), as well as backup missions ready to be activated if some satellites were to die unexpectedly.

The past decade was also transformational for satellite altimetry in terms of technology. The CRYOSAT-2 satellite demonstrated that delay-Doppler (Synthetic Aperture Radar Mode, or SARM) was beneficial over the ocean and not just the cryosphere. It also allowed the research community to explore benefits of cross-track interferometry in a very narrow swath. The demonstration carried out with CRYOSAT-2 culminated in the activation of SAR mode for the global ocean with Sentinel-3A and 3B. As Doppler altimetry processors were improving (e.g. Moreau et al., 2021), Delay-Doppler altimetry made another leap with Sentinel-6MF and its so-called "interleaved" mode (e.g. concurrent Jason-class and SARM products, fully focused SAR for coastal applications). In parallel, Ka-band was successfully validated with the SARAL mission. Verron et al (2021) reports smaller footprint and less noise. SARAL paved the way for the CRISTAL missions combining Ku-band and Ka-band, as well as for SWOT's Ka-band radar interferometer (KaRIn).

Furthermore, in comparison with these incremental improvements (better performance and better resolution), the most impactful change happened in 2023 with SWOT's first large-swath altimeter (Fu et al., 2024). It demonstrated not only a major gain in coverage (equivalent to 50 altimeters-worth of pixels) but also an order of magnitude gained in precision with respect to nadir altimetry. This new technology proved directly useful for ocean prediction (e.g. Tchonang et al., 2021; Benkiran et al. 2025) as well as indirectly beneficial through research: new mean sea surface models, new bathymetry (e.g. Yu et al., 2024), better understanding of internal tides (e.g. Tchilibou et al., 2024), better coverage for new barotropic tides, small/sub mesoscale processes, etc. This impactful change comes with a series of challenges ranging from the validation of this new type of measurement to the blending of very different technologies in a consistent system (Dibarboure et al., 2024). To that extent, SWOT is expected to unlock various improvements that will be beneficial to older nadir altimeters (e.g. reprocessing with better tide model or mean sea surface models). At the time of this writing, swath altimetry is also the main candidate technology for the next generation of Sentinel-3 missions (S3NG) from the Copernicus Programme, while Doppler and Ka-band are the baselines for other operational missions.

## 2.2 Evolution of the DUACS altimeter data processing system

Since the early 2000's, DUACS (data unification and altimeter combination system) has served as the primary multi-mission altimeter data center providing high-quality, near-real-time, and delayed-mode global and regional altimeter products to the scientific community and operational oceanography. Main processing steps are product homogenization, data editing, orbit

error correction, reduction of long wavelength errors, and production of along-track and maps of sea level anomalies (SLA) and absolute dynamic topographies. A comprehensive description of DUACS is given in Dibarboure et al. (2011) and Pujol et al. (2016). Since 2014, the Copernicus Marine Sea Level and Wave Thematic Assembly Centers have been part of the DUACS altimeter data processing system.

The DUACS sea level system has been designed to address three primary objectives: the provision of L3 products dedicated to assimilation into numerical models and the restitution of mesoscale signals, notably for the Copernicus Marine Service (CMEMS), and the restitution of climatic scales, notably for the Copernicus Climate Change Service (C3S).

Several factors contribute to the quality and relevance of the products: the number of altimeters considered, the choice of retracking standards, the various geophysical and environmental corrections applied to the altimeter measurements, the choice of L3 and L4 processing algorithms (e.g. improved selection of valid data, denoising, new mapping methods, addition of variables of interest, etc.) and the physical content of the products. The main evolution of processing capabilities and level 2, 3 and 4 data products that are relevant to operational oceanography and its applications are summarized below.

Over the years, several iterations of reprocessing of L2 input products have been carried out, to take into account new retracking, instrumental corrections or new geophysical and environmental correction solutions. Similarly, a complete reprocessing of the L3/L4 series has been carried out. A detailed analysis of the new products and standards available is carried out to select those that will best meet the objectives for Copernicus Marine and Climate Change services (e.g. Kocha et al., 2024). Similarly, the L3/L4 processing algorithms are reviewed and adjusted for each reprocessing, in particular about parameter evolution and data interpolation methods. Four versions of the reprocessed series have been delivered since 2010. They are identified as "DT-2014" (Pujol et al., 2016), "DT-2108" (Taburet et al., 2019), "DT-2021" (Faugère et al., 2022) and "DT-2024" (Pujol et al., 2024a). These were followed by an upgrade of NRT production to bring it into line with the same standards and treatments. The various product versions have reduced SLA errors in the L4 mesoscale product (wavelengths between 65 and 500 km) by 35% in areas of high variability, compared with the previous version (DT-2010) (Pujol et al., 2024b). Improvements to ocean tide models (Carrere et al., 2016; 2023) and Mean Sea Surface (MSS) (e.g. Schaeffer et al., 2012; 2022; Pujol et al., 2018; Laloue et al., 2025) represent an important part of the improvement in data quality, particularly in coastal areas (Kocha et al., 2024; 2025). Another area of improvement is the refinement of the mapping processing, with the initial evolution of the Optimal Interpolation (OI) parameters at global and regional scales (Pujol et al., 2016; Taburet et al., 2019; Faugère et al., 2022) and subsequently the implementation of a new Multiscale Inversion of Ocean Surface Topography (MIOST) mapping process (MIOST; Ubelmann et al., 2021; Ballarotta et al., 2023) that allows for the definition of covariances associated with various surface dynamical processes (rather than the single-scale covariance function used in the OI methodology). In the current version of the L4 product (i.e. DT-2024), the average spatial resolution of the global gridded SLA is around 180 km at mid-latitudes, with larger values near the equator (~500 km)

and finer values at the poles (~100 km). Compared with the previous DT-2014 version (discussed in Ballarotta et al, 2019), the effective spatial resolution has been improved by almost 10% at mid-latitudes; the bulk of the improvements being made by changes in the DT-2024 version with a significant improvement induced by the MIOST mapping method. Note that, as mapping methods are designed to map anomalies relative to a zero mean, data selected for a given grid point estimation are recentered before the L4 mapping. Mean bias is then reinjected in the solution. The integrity of large-scale spatial and temporal signals (e.g., ocean level trends) is thus ensured.

By exploiting the tandem phases between two successive reference missions, it is possible to correct the sea level biases observed between two missions on a regional scale. This correction, essential for sea level rise monitoring, has been re-estimated and refined over the years (Kocha et al., 2024; 2025; Cadier et al., 2025). In parallel, certain measurement anomalies, which have a substantial impact on sea level rise monitoring, are corrected at the L3 level. One of the main corrections consists in an empirical correction of the TOPEX-A measurement drift, estimated by comparison with in-situ measurements (Ablain et al., 2017). Wave-dependent biases in Sentinel-6A HR and LR measurements have also been empirically estimated for correction at L3 level, while the origin of these anomalies is now corrected at L2 level (Kocha et al., 2024). Finally, the levels of uncertainty in the estimation of the mean sea level trend on a global and regional scale are regularly reviewed (Ablain et al., 2009; 2019; Prandi et al., 2021; Guerou et al., 2022). All these processes help to reduce uncertainty levels and better highlight the acceleration of sea level rise, as discussed for example in Von Schuckmann et al. (2024).

L3 sea level products used for data assimilation have evolved over the years. Since 2015, the L3 product has included several variables required for optimum assimilation into ocean prediction systems. The L3 product now contains not only the sea level anomalies and absolute dynamic heights, but also the various geophysical corrections applied to the data (like ocean tide, dynamic atmospheric correction, and long wavelength errors). This allows modelers to reinject this signal into the assimilated data to better match its physical content to that of the model, and thus better constrain their model (see section 3).

Since November 2023, a major milestone in the evolution of DUACS products has also been reached, with an increase in product resolution. The NRT sea level system now takes the 20 Hz initial resolution data as input, rather than the 1 Hz resolution previously used. This was made possible by the advances in altimeter techniques and processing, such as the Synthetic Aperture Radar (SAR) measurement mode, which allows for a significant reduction in measurement noise (e.g., by about 30% as underlined by Raynal et al., 2018). Additionally, specific empirical noise mitigation corrections enable the reduction of measurement noise in conventional LRM measurements (Tran et al., 2021). This improves the ability to observe coastal and small mesoscale signals (Pujol et al., 2023). Unlike the historical 1 Hz processing, the 20 Hz noise filtering processing is reviewed to optimally filter the measurement at the regional scale, taking into account the specific observing

capability of each mission (up to 40 km for Sentinel-6A i.e. locally up to one third compared with the conventional 1 Hz product). The L3 product defined with 5 Hz sampling, i.e., slightly more than 1 km between two consecutive measurements, is a good compromise between the sampling needs and observing capabilities. The development of this product was motivated by the necessity to address the increasing resolution of ocean models (see discussions in sections 2.4 and 3 and in Pujol et al., 2023).

The DUACS system has also been improved to reduce real-time data availability timeliness. In 2009, modifications were implemented in the production of L3 DUACS data to account for the mesoscale content of fast delivery upstream L2 products (OGDR/NTR). The availability of these data is limited to a few hours after measurement, yet they enable operational applications to retrieve useful information on the last day or two of measurement, depending on the mission. This enhancement has been demonstrated to reduce errors in multi-mission products by 20% (Dibarboure et al., 2009). Secondly, the system has undergone continuous upgrades to facilitate the rapid processing of all data. Despite the increase in the number of altimetry missions to be processed (3 to 4 missions in 2015, compared with 8 in 2024) and the augmented data volume associated with the enhanced resolution, the optimal data availability time was nearly 96% in 2015. It has persisted above 99% since 2018 (Table 1).

| Timeliness on 2015-2021 period | | | | | | | |
|---|---|---|---|---|---|---|---|
| Year | 2015 | 2016 | 2017 | 2018 | 2019 | 2020 | 2021 |
| Timeliness | 96,31% | 97,63% | 98,35% | 100,00% | 99,85% | 99,51% | 99,45% |
| Mean Phase 1/2 | 97,43% | | | 99,70% | | | |
| Mean Copernicus1 | 98,73% | | | | | | |

Table 1: Copernicus Marine Sea Level **Thematic Assembly Center** products timeliness over the 2015-2021 period. The timeliness criterion represents the percentage of availability of the daily production over a quarter. The mean timeliness for Copernicus 1 phase 1 (2015-2017), Copernicus 1 phase 2 (2018-2021) and Copernicus 1 (2015-2021) are also given.

Future sea level products

The increased resolution of the products, with data processed directly at 20 Hz, also enables us to prepare for future DUACS products, particularly in polar and coastal regions where the altimeter signal and the geophysical corrections applied to it are contaminated by the presence of ice and/or land. Since the 2010s, several studies have highlighted the need to specifically process data from ice fracture zones to extract surface level measurements in these areas, while ensuring signal continuity with the open ocean (e.g., Prandi et al., 2021b). These treatments have recently been taken into account by space agencies for the generation of future L2 products. More recently, several studies have demonstrated the feasibility of multi-mission L3/L4 treatments in polar zones (Prandi et al., 2021b; Auger et al., 2022; Veillard et al., 2024). Such products are likely to become part of the Copernicus Marine Service catalogue in the coming years. Similarly, the quality of altimetry measurements in

coastal areas requires the application of specifically selected corrections (e.g. Birol et al., 2024) and is greatly enhanced by the use of innovative techniques such as SAR (Vignudelli et al., 2011).

Finally, the launch of SWOT-KaRIn in December 2022 (see above), with its exceptional spatial sampling and its ability to improve the various fields involved in processing altimetry measurements, should greatly enhance all DUACS sea level products in the near future. First demonstration L3 and L4 products are already available (Dibarboure et al., 2024; Ballarotta et al., 2024).

## 2.3 Evolution of MDTs

One fundamental requirement for assimilating SLA in ocean models is accurate knowledge of the mean dynamic topography (MDT) (Le Traon et al., 2017). MDT is the difference between the mean sea surface (MSS) and the Earth geoid and as such it constitutes, to within a constant and with some remaining uncertainties, the missing link between the observed sea level anomalies and the modelled sea surface heights. Thanks to more than 30 years of altimetry observations and dedicated geodetic phases and/or geodetic missions, the MSS is known with centimeter accuracy at spatial scales down to a few km (see Schaeffer et al., 2023 for a recent update). Thanks to the GOCE and GRACE gravity missions, knowledge of the geoid at scales of 100-150 km has greatly improved (Flechtner et al, 2014), so that the ocean MDT is now resolved at those scales with centimetre accuracy. However, the true ocean MDT contains scales shorter than 100-150 km. To compute higher resolution MDT, gravity mission data can be combined with altimetry and in-situ measurements such as temperature and salinity profiles from the Argo array and velocity measurement from drifting buoys (e. g. Rio and Hernandez, 2004). Over the past years, a series of new MDTs has been produced from global to regional scales. They all benefit from improved input data, such as a better quality geoid (e.g from EGM96 (Lemoine et al., 1998) to GOCO06S (Kvas et al., 2021)) and MSS data (e.g from MSS CNES_CLS_01 (Hernandez et al., 2002) to MSS_CNES_CLS_2022 (Schaeffer et al., 2023)), the availability of more in situ data accumulated over time, and the evolution of processing methods. The main changes are: the inclusion of T/S profile data at different reference depths, the estimation of mean currents in the equatorial band (MDT CNES_CLS_2009 (Rio et al., 2011)), improved drifter data processing  (MDT CNES_CLS_2013 (Rio et al., 2014)), changing the altimeter reference period to center the MDT on the 20-year period [1993, 2012] (MDT CNES_CLS_2013), and using surface current measurements from High Frequency Radars (MDT CNES_CLS_2022) (Caballero et al., 2020; Jousset et al., 2023).

The Mercator Ocean analysis and forecasting systems use «hybrid» MDTs (e.g. Hamon et al., 2019; Lellouche et al., 2021) where iterative corrections of the first guess of an MDT derived from gravimetry and in-situ observations are performed using high-resolution reanalysis system outputs, updates to the GOCE geoid and an improved post-glacial rebound (also called glacial isostatic adjustment). The main advantage is to reduce MDT errors in coastal and high latitude regions (close to sea ice), ensure the consistency of the MDT with model dynamics and obtain an MDT at the model resolution.

## 2.4 Operational oceanography and ocean prediction

Major improvements of ocean prediction systems have occurred from the GODAE (Global Data Assimilation Experiment) demonstration (Smith and Lefebvre, 1997) in the 2000s to fully operational systems that now serve a large range of applications in the 2020s. Ocean modelling and data assimilation systems operationally assimilate in-situ and satellite data to provide regular and systematic reference information on the physical state, variability and dynamics of the ocean and marine biogeochemistry from global to coastal scales (e.g. Chassignet et al., 2018; Le Traon et al., 2019; Heimbach et al., 2019;

Fennel et al., 2019). Products serve a wide range of applications and downstream services (e.g. Bell et al., 2015; Le Traon et al., 2019; Schiller et al., 2019; Le Traon et al., 2021).

The French contribution to GODAE began in the late 90s with the development of Mercator Ocean center, the Coriolis partnership for the in-situ component (including Argo) and the development of the Jason series with CNES (French Space

Agency). Mercator Ocean issued its first operational bulletin of the Atlantic in 2001. 2005 was a major milestone with the start of the first global ocean prediction system at a 1/4° resolution (Drevillon et al., 2008). Since then, Mercator Ocean analysis and forecasting global systems have continuously evolved (Lellouche et al. 2013; Lellouche et al., 2018; Lellouche et al., 2023) in the framework of the MyOcean European research projects and the Copernicus Marine Service (Fig. 3).

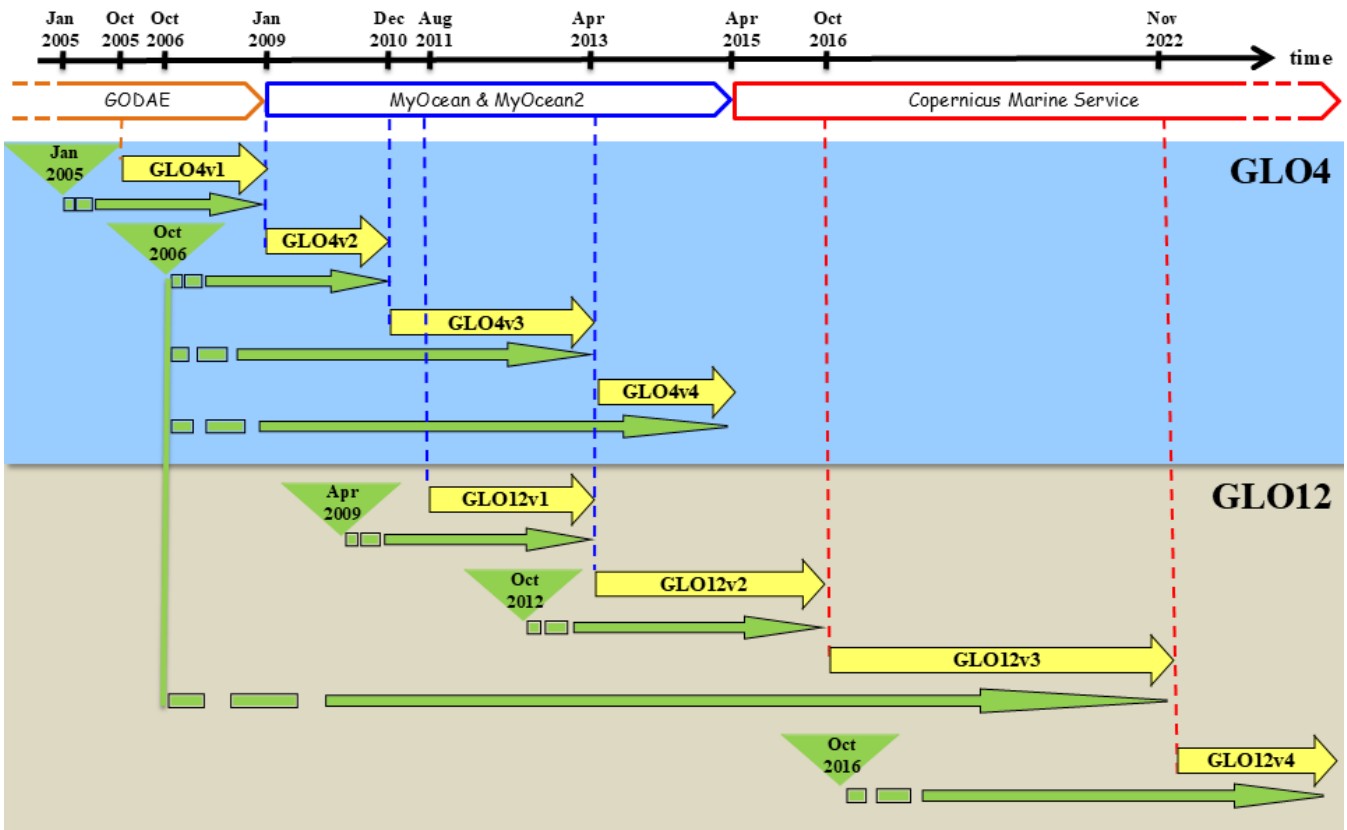

**Figure 3: Timeline of the Mercator Ocean global analysis and forecasting systems for the various milestones in the past: GODAE, MyOcean projects and the current Copernicus Marine Service. Real-time productions are in yellow with the reference of the Mercator Ocean system. Available Mercator Ocean simulations are in green, including the catchup to real time. Global systems at 1/4° (1/12°) are referred to as GLO4 (GLO12).**

Since its creation, Mercator Ocean has adopted a strategy of incremental improvement of its assimilation schemes. The initial version consisted of a reduced-order optimal interpolation scheme which had proved its efficiency for the assimilation of altimetry data (Ferry et al., 2007). The SLA increment was split into baroclinic and barotropic contributions using statistical information from the system. The barotropic component was then converted into an increment of horizontal velocity and barotropic stream function of the model, while the baroclinic part was converted into vertical corrections of
temperature and salinity by lifting or lowering of isopycnals with a method derived from Cooper and Haines (1996) and then converted into geostrophic current increments. This scheme was then improved to be able to assimilate satellite observations (altimetry and surface temperature) together with in situ observations (XBT, profilers) in a multivariate/multi-data version for regional or basin-scale operational systems. The second generation of the Mercator Assimilation System is based on a SEEK (Singular Evolutive Extended Kalman filter) analysis kernel (Brasseur and Verron, 2006). Compared with the
previous version, its main feature is its improved representation of the background error by a 3D multivariate error subspace, which has helped to overcome some of the limitations identified in the previous scheme, such as its unsuitability for shallow areas and tropical oceans, and the difficulty of optimally controlling the surface layer. The formulation of the SEEK analysis kernel is also characterized by its greater efficiency in processing large quantities of observations. The first version of this tool also included two major parameterizations: a local approach for the analysis and an adaptive scheme for the background
variance improving the consistency of the error statistics for the filter input. The initial version of this assimilation scheme was deployed progressively from 2007 onwards on all Mercator Ocean forecasting and reanalysis systems. It was updated significantly in 2010-2011 (Lellouche et al., 2013) and now forms the main data assimilation tool of the current Mercator Ocean systems.

Mercator Ocean now operates a global real-time monitoring and forecasting system at 1/12°, hereafter referred to as GLO12v4 (Lellouche et al., 2023) and produces and upgrades regularly a global ocean reanalysis at 1/12°, hereafter referred to as GLORYS12 (Lellouche et al., 2021). Both systems are based on a NEMO model configuration and assimilate along-track altimetry observations from the Copernicus Marine Sea Level Thematic Assembly Center (TAC), satellite sea surface temperature, in situ temperature and salinity vertical profiles and satellite sea-ice concentration. A regional system (IBI / real
time and IBIRYS / reanalysis) that includes tidal forcing covers the Northeast Atlantic with a higher (1/36°) resolution. All these integrating systems use a 7-day assimilation window, and the forecasting systems deliver daily 10-day forecasts. The increments coming from the analysis are applied progressively using the incremental analysis update (IAU) method (Benkiran and Greiner, 2008) which makes it possible to avoid model shock every week due to the imbalance between the analysis increments and the model physics, and results in an optimal and continuous model trajectory. Moreover, all these

system evolutions have required a very significant increase in the computing and storage capacities needed to carry out the developments, and to produce real-time forecasts and reanalyses. Computing and storage capacities remain a major challenge but allow for an increase in resolution and the development of ensemble approaches for the global system (see section 5).

The main evolution of Mercator Ocean global operational ocean prediction systems from 2005 to 2024 is summarized in Table 2. The evolution of the performance of the global system at 1/12° (from v1 to v4) (Fig. 4) shows a gradual improvement from one version to the next. This highlights the system advances in error reduction compared to altimetric observations.

| Mercator Ocean system reference | Resolution | Model | Assimilation | Assimilated observations |
|---|---|---|---|---|
| GLO4v1 *(PSY3 in Drevillon et al 2008)* | Horizontal: 1/4° Vertical: 46 levels | ORCA025 OPA 8.2 24 h atmospheric forcing Relaxation towards SST | Reduced Order Optimal Interpolation | L3 SLA |
| GLO4v2 | Horizontal: 1/4° Vertical: **50 levels** | ORCA025 **NEMO 1.09 LIM2 ice model, Bulk CLIO** 24 h atmospheric forcing | **SAM (SEEK)** | L3 SLA **L4 AVHRR SST** **In-situ T/S vertical profiles** |
| GLO4v3 | Horizontal: 1/4° Vertical: 50 levels | ORCA025 **NEMO 3.1 LIM2 EVP ice model, Bulk CORE** **3 h atmospheric forcing** | SAM (SEEK) **IAU** **3D-VAR T and S bias correction** | L3 SLA L4 AVHRR SST In situ T/S vertical profiles |
| GLO4v4 | Horizontal: 1/4° Vertical: 50 levels | ORCA025 NEMO 3.1 LIM2 EVP ice model, Bulk CORE 3 h atmospheric forcing **New parameterization of vertical mixing** **Adding seasonal cycle for surface mass budget** | SAM (SEEK) IAU 3D-VAR T and S bias correction **Obs. errors higher near the coast and on shelves** | L3 SLA L4 AVHRR SST In-situ T/S vertical profiles **MDT "CNES-CLS09" adjusted** **In-situ Sea Mammals T/S vertical profiles** |
| GLO12v1 *(PSY4 in Lellouche et al 2013)* | Horizontal: 1/12° Vertical: 50 levels | ORCA12 NEMO 1.09 LIM2, Bulk CLIO 24 h atmospheric forcing | SAM (SEEK) IAU | L3 SLA L4 AVHRR SST In-situ T/S vertical profiles |
| GLO12v2 | Horizontal: 1/12° Vertical: 50 levels | ORCA12 **NEMO 3.1 LIM2 EVP ice model, Bulk CORE** **3 h atmospheric forcing** **New parameterization of vertical mixing** **Large scale correction to the downward radiative and precipitation fluxes** **Adding seasonal cycle for surface mass budget** | SAM (SEEK) IAU **3D-VAR bias correction** **Obs. errors higher near the coast and on shelves** | L3 SLA L4 AVHRR SST In-situ T/S vertical profiles **MDT "CNES-CLS09" adjusted** **In-situ Sea Mammals T/S vertical profiles** |
| GLO12v3 | Horizontal: 1/12° Vertical: 50 levels | ORCA12 NEMO 3.1 LIM2 EVP, Bulk CORE 3 h atmospheric forcing New parameterization of | SAM (SEEK) IAU 3D-VAR bias correction **Adaptive tuning of** | L3 SLA **L4 OSTIA** SST In-situ T/S vertical profiles **MDT adjusted based on** |

| | | | | |
|---|---|---|---|---|
| | | vertical mixing<br>Adding seasonal cycle for surface mass budget<br>**50% of model surface currents used for surface momentum fluxes**<br>**New correction of precipitations using satellite data, correction of the downward radiative fluxes removed** | observation errors for SLA and SST<br>**New 3D observation errors files for assimilation of in situ profiles** | CNES-CLS13<br>In-situ Sea Mammals T/S vertical profiles<br>**L4 Sea Ice Concentration**<br>WOA13v2 climatology (temperature and salinity) below 2000 m (assimilation using a non-Gaussian error at depth) |
| GLO12v4 | Horizontal: 1/12°<br>Vertical: 50 levels | ORCA12 **NEMO 3.6**<br>**LIM3 EVP multicategory ice model**, Bulk CORE<br>**1 h atmospheric forcing**<br>**New parameterization of vertical mixing**<br>Adding seasonal cycle for surface mass budget<br>**Use of satellite-based monthly estimates of the global mean sea level to better constrain the ocean mass**<br>Correction of precipitations using satellite data; correction of the downward radiative fluxes removed | SAM (SEEK) + **4D extension of the data assimilation scheme**<br>**Improved parametrization of the model error covariance**<br>**Assimilation of "super-observations"**<br>**IAU**<br>**New 3D-VAR bias correction**<br>**Reduction of velocity increments under 2000 m** | L3 SLA<br>**L3 ODYSSEA** SST<br>In-situ T/S vertical profiles<br>**New adjusted MDT**<br>In-situ Sea Mammals T/S vertical profiles<br>L4 Sea Ice Concentration<br>**EN4** climatology (temperature and salinity) below 2000 m **(only used for bias correction)** |


**Table 2: Evolution of the Mercator Ocean global ocean prediction systems at 1/4° (GLO4) and 1/12° (GLO12) since 2005. In bold, the major upgrades with respect to the previous version. Available and operational production periods are described in Figure 3.**

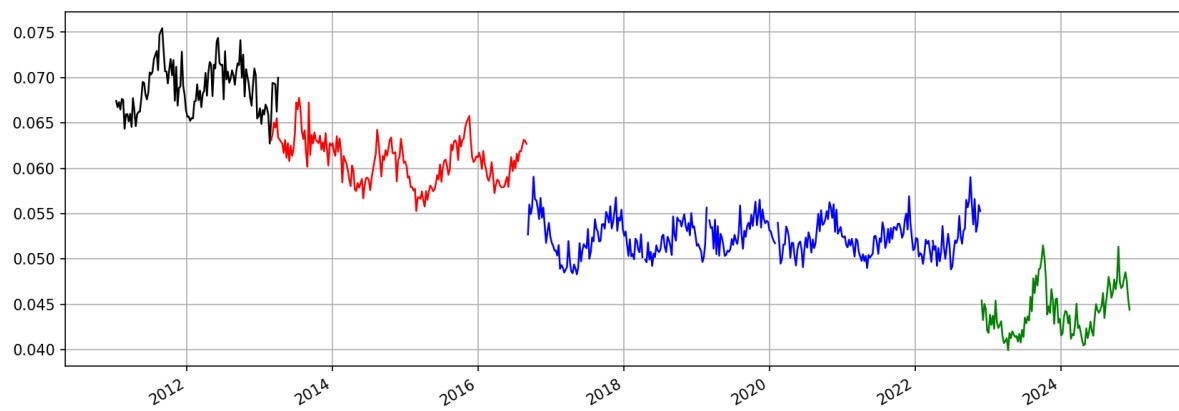


**Figure 4: Evolution of the performance of the global Mercator Ocean high-resolution prediction system since 2011 (RMS of the difference between sea level modelled fields and altimeter observations for the four versions of the Mercator Ocean 1/12° system). The colour indicates the version of the system (GLO12v1 in black, GLO12v2 in red, GLO12v3 in blue and GLO12v4 in green) and units are meters. From Pinardi et al. (2024).**

From 2005, most of the ocean prediction developments including the ones of Mercator Ocean have been carried out in Europe in the framework of the Copernicus Marine and its precursors R&D projects (Mersea and MyOcean) (e.g. Johannessen et al., 2006; Le Traon et al., 2019). The Copernicus Marine Service (see Le Traon et al., 2021 for a review of past achievements) now provides the EU with a world-leading, reference operational oceanography service on the physical, biogeochemical ocean and sea-ice state of the world ocean and EU regional seas. All Copernicus Marine Monitoring and Forecasting Centers (MFCs) provide quality-controlled reanalyses, analyses and 10-day forecasts and assimilate altimeter data (sea level and significant wave height) from the Copernicus Marine Sea Level and Wave Thematic Assembly Centers. Copernicus Marine directly contributes to EU's marine and maritime-related policies and supports applications dealing with maritime safety, sustainable use of marine resources, healthy waters, informing coastal and marine hazard services, ocean climate services, protecting marine biodiversity (see https://marine.copernicus.eu/services/use-cases). More than 90,000 expert services and users worldwide are connected to the service.

## 3 Impact of altimetry on ocean prediction systems

### 3.1 Methods to assess impact of altimetry for ocean prediction

The monitoring of the impact of observations is a central function of an ocean prediction center. This is done through Observing System Evaluations (OSEs) and Observing System Simulation Experiments (OSSEs) (e.g. Fuji et al., 2019; Tchonang et al., 2021). OSEs allow assessing the impact of an existing data set on the performances of an ocean prediction system (by withholding observations). OSSEs provide a comprehensive virtual framework to help designing new observing systems, evaluate their different configurations, and to perform preparatory data assimilation work. Alternative and complementary approaches (e.g. computation of Forecast Sensitivity-based Observation Impacts) (see Fuji et al., 2019) are also existing but are less commonly used in operational oceanography mostly because they require an adjoint or an ensemble data assimilation scheme (e.g. Drake et al., 2023).

From OSEs and OSSEs, different metrics are used to compare assimilation results (analyses and forecasts) with a reference (e.g. a truth run for OSSEs, independent/non assimilated data sets or a reference run for OSEs). This includes calculating error variances for full fields (e.g. sea surface height, currents, temperature, salinity) or for special spectral bands, F/W spectra of errors (for OSSEs), normalized spectra, coherence analysis, resolved scales, Lagrangian analyses (e.g. D'Addezio et al., 2019; Jacobs et al., 2021; Tchonang et al., 2021). In addition to verifying the efficiency of the assimilation scheme to reduce the misfit to the assimilated observations, those diagnostics highlight how different unobserved regions and variables and which spatial and temporal scales are improved thanks to the additional information of observations (e.g. Gasparin et al. 2023). Lagrangian analyses provide more insights of the benefits for applications related to the drifts of tracers such as pollutants or plastic debris.

## 3.2 Main results and findings

High resolution from multiple altimeters is required to adequately represent ocean eddies and associated currents in models. Both Observing System Evaluations (OSEs) (e.g. Hamon et al., 2019) and Observing System Simulation Experiments (OSSEs) (e.g. Verrier et al., 2017) demonstrate the major contribution of altimetry. The new generation of nadir altimeters now provides enhanced capability, thanks to a Synthetic Aperture Radar (SAR) mode that reduces measurement noise. Verrier et al. (2018) demonstrated that the use of SAR multiple altimeter missions with high-resolution models will improve the capability of ocean analysis and forecasting systems. Gasparin at al. (2023) highlighted the complementary role of satellite and in situ observations to constrain the large scale and mesoscale variability and the need for high-resolution altimeter observations to constrain the mesoscale.

At least four altimeters are required to constrain modeling and data assimilation systems. This is particularly true with high-resolution data assimilation systems. As an illustration, Fig. 5 shows the impact of the assimilation of a fourth altimeter (S3-A) data in the GLO12v3 system (International Altimetry Team, 2021). Two runs were performed over a one-month period (May 2017) (International Altimetry Team, 2021): one with the assimilation of Jason-3, SARAL/AltiKa and CryoSat-2 and one with the assimilation of S3A, Jason-3, SARAL/AltiKa and CryoSat-2. Adding a fourth altimeter allowed reducing the variance of 7-day forecast errors by about 10%. Forecast errors are reduced up to 7 cm rms in Western Boundary Currents. Adding more altimeters has always a positive effect on system performance, although the effect decreases as more altimeters are added. For example, analysis error variances (estimated by comparison with a non-assimilated nadir altimeter) are typically reduced by 10-15% when moving from 3 to 6 altimeters.

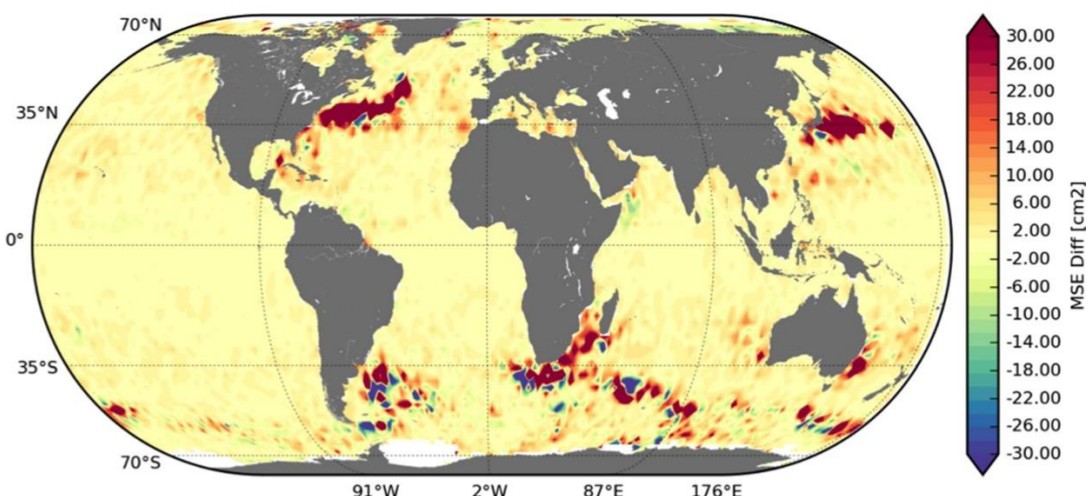

**Figure 5: Forecast error reduction due to the assimilation of Sentinel-3A data in addition to Jason-3, SARAL/AltiKa and CryoSat-2 in the Mercator Ocean 1/12° global system.**

Accurate knowledge of the Mean Dynamic Topography (MDT) is a fundamental element for assimilation into ocean models. Hamon et al. (2019) showed that, in terms of impact on sea level, assimilating an updated release of the MDT is comparable to assimilating a fourth altimeter. Gasparin et al. (2021) documented the impact of MDT errors on equatorial dynamics and BGC models. Thanks to new and improved altimetric, in-situ and gravimetric (GRACE and GOCE satellite missions) data,
MDTs are regularly updated, leading to considerable improvements in both forecasts and analyses (see section 2).

The 1 Hz L3 along-track SLA Copernicus Marine Sea Level product has been recently updated with a new 5 Hz L3 product (see section 2.2). To highlight the impact of these 5 Hz (L3) altimeter data on the global forecast system, an OSE was carried out with the GLO12v4 system. Two simulations were carried out over a period of 5 months, one with 1 Hz SLA data and the
415 other with the 5 Hz data. The assimilation of SLA data at 5 Hz with its improved filtering approaches was shown to provide a significant improvement in the analyses and forecasts (reduction of variance errors by a few %). It will be implemented in the operational production end of 2025.

Today, seven altimeters are assimilated routinely in the current Mercator Ocean 1/12° operational system GLO12v4. Figure
6 shows the time evolution over the 2017-2024 period of the SLA innovations (differences between observations and background model first trajectory) for the different assimilated altimeters. This misfit is consistent in time and across different altimeters. The mean (over the global ocean) innovation shows that the system is lower than the observations by around 1 to 2 mm on average and the global RMS difference rarely exceeds 5.5 cm. The seasonal cycle in the innovation RMS is also observed in the SLA variability and is most likely related to the evolution of the altimeter coverage due to the
ice cover in the Arctic.

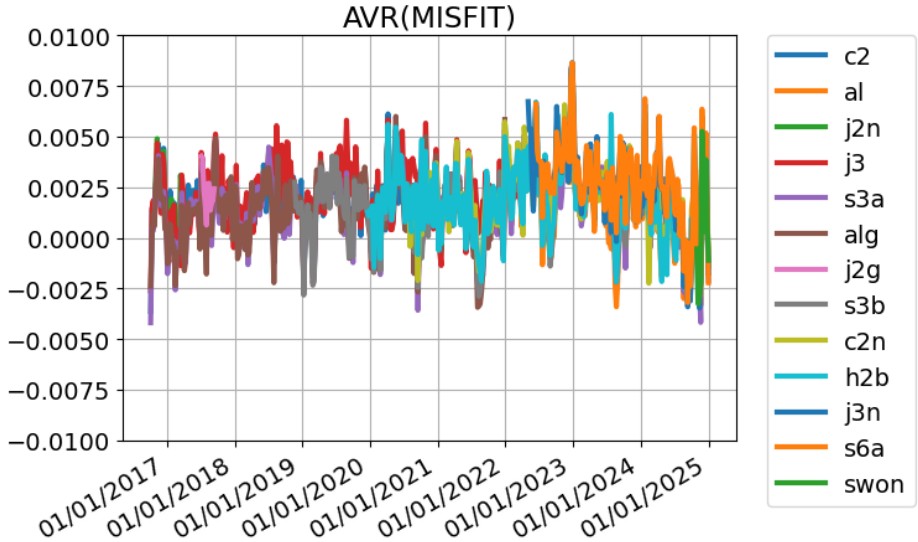

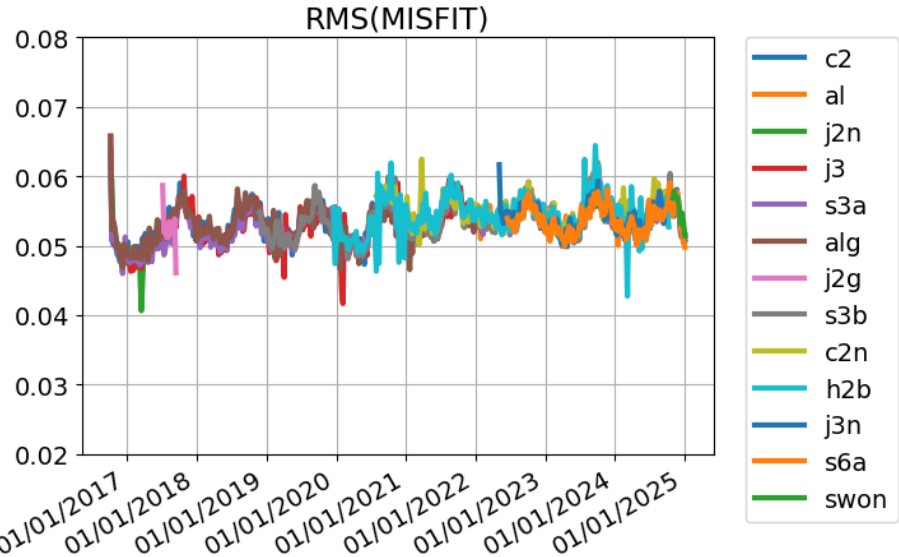

**Figure 6: Temporal evolution of the global mean SLA innovation (observation – background model first trajectory) (top panel) and RMS of this misfit (below panel) for the different altimeter data sets assimilated in GLO12v4: c2 (Cryosat-2), al (Alti-Ka), j2n (Jason-2 tandem phase), j3 (Jason-3), s3a (Sentinel 3a), alg (Alti-Ka geodetic phase), j2g (Jason-2 geodetic phase), s3b (Sentinel 3b), c2n (Cryosat-2), h2b (Hy 2b), j3n (Jason-3 tandem phase), s6a (sentinel 6a), swon (Swot nadir). Units are meters.**

## 4 The development of swath altimetry and its role for ocean prediction

Operational oceanography requirements (3 altimeters in addition to a reference altimeter mission in optimized orbits) for satellite altimetry were defined almost 20 years ago (e.g. see Johannessen et al., 2006). Since then, the ocean analysis and forecasting systems have strongly evolved and so has our understanding of the impact of altimeter observations in models. A constellation of 4 altimeters cannot resolve wavelengths below 150-200 km and time scales of 10 to 20 days. As discussed in Jacobs et al. (2021) and Jacobs et al. (2023), observing systems resolve a range of scales that are larger than scales models represent. Small-scale model features (e.g. small mesoscale and submesoscale eddies and frontal instabilities) are unconstrained and are not in the correct location leading to a double penalty effect (Jacobs et al., 2021). As observation resolution increases the separation between constrained and unconstrained scales should move to smaller scales.

Much higher space/time resolution is needed in the post 2030 period due to the increase of global and regional model resolution. Beyond 2028, the resolution of Mercator Ocean /Copernicus Marine models is expected to increase by a factor of 3 (e.g. global 1/36°, regional 1/108°). The resolution of current altimeter products is not able to constrain the smallest scales of such high-resolution models. The limitation is emphasized in coastal regions where the combined effects of mesoscale, wind, bathymetry and tides can generate very complex small scale and high frequency effects.

The primary focus for an altimeter constellation for operational oceanography should be the observation of ocean dynamics from large scales to mesoscale as well as an improvement in the coastal ocean. The objective proposed for the Copernicus Marine Service (MOi, 2024) would be to resolve wavelengths larger than 50 km every 5 days. Model simulations and in-situ data indicate that the scale where balanced and unbalanced motions have the same influence on sea level generally ranges from 30 to 80 km, with seasonal and geographical variability (e.g. Qiu, 2018). Given their space/time scales, these motions are unlikely to be assimilated and propagated in ocean models. To that extent, it is relevant to limit the main requirement to wavelengths of 50 km and 5 days. This limit is already quite challenging but credible from an implementation point of view and consistent with the effective resolution of global and regional models (5 to 10 times larger than the model grid, see Soufflet et al., 2016). The coverage and revisit requirements can be guaranteed with several swath altimeters and an altimeter nadir constellation.

The development of swath altimetry demonstrated with the SWOT mission opens up very promising perspectives to improve our ability to monitor and forecast the smaller space and time scales (e.g. d'Addezio et al., 2019; Tchonang et al., 2021). Through a series of OSSEs, Tchonang et al. (2021) demonstrated that SWOT data could be readily assimilated in a global high resolution (1/12°) analysis and forecasting system with a positive impact everywhere and very good performance. Adding SWOT observations to those of three nadir altimeters globally reduced the variance of sea level and surface velocities analyses and forecasts by about 30% and 20% respectively. Assimilation of real SWOT data in the Mercator Ocean global 1/12° model shows promising initial results (Benkiran al., 2025). Compared to results from 3 nadir altimeters, the addition of SWOT improves the analyses and forecasts by 15 to 20%. Future improvements in the data assimilation system (model and observation error characterization, scale separation) should lead to better results. The main limitation of SWOT is, however, related to its 21-day repeat period. In the long run, flying a constellation of several swath altimeters and nadir altimeters would thus be highly beneficial to further improve performance, in particular, for the small space and time scales (e.g. Benkiran et al., 2022; Benkiran et al., 2024).

## 5 Prospects for the next decade

Prospects for the joint development of satellite altimetry and operational oceanography for the next decade look promising. The altimeter constellation will be expanded (e.g. CRISTAL, S3C&D, S6B&C and S6 NG, S3NG, HY2) (see Fig. 1). The continuation of the reference mission with S6B&C and S6 NG is essential to ensure the homogenization of the constellation and the monitoring of climate signals. Polar regions will be better observed thanks to the CRISTAL missions. From 2032, the S3NG swath altimetry two-satellite constellation will play a major role to constrain the future high resolution ocean prediction systems.

DUACS/Copernicus Marine Sea Level TAC altimeter products will continue to be improved (see section 2) to optimize their use and impact for operational oceanography: new geophysical corrections (e.g. barotropic and internal tides), better resolution, improved retracking and filtering / noise reduction approaches, improved quality for coastal areas, new products in sea ice regions, reduced timeliness for data delivery (e.g. < 3 hours). Developments to improve the quality and resolution of MDTs from global to coastal scales will be carried out.

Ocean prediction systems will evolve from global to coastal scales with resolution ranging from a few km at global scale to hundreds of meters at coastal scales, with improved data assimilation schemes, inclusion of new processes and improved coupling (wave, atmosphere), development of ensemble approaches and increased use of Artificial Intelligence (AI). AI is a rapidly evolving field and has a major potential for ocean and sea ice forecasting and to improve analysis and forecasting systems (e.g. model emulation, subgrid scale parametrisation, calibration and bias correction) (e.g. Heimbach et al., 2024). By providing dynamic and interactive cloud-based platforms to model and predict the ocean, explore various scenarios, assess impacts to make informed decisions, ocean digital twins are also transformative technologies that will offer new perspectives for the development of operational oceanography.

As far as MOi systems are concerned, an ensemble-based analysis and forecasting global system at 1/4° (with 1/12° nested domains using the AGRIF (Adaptive Grid Refinement In Fortran) tool in high eddy energy regions) to complement the deterministic system at 1/12° will be put in operation in the coming years. This system will include tidal and atmospheric pressure forcings and will provide uncertainty estimates for the higher resolution deterministic system and longer-term forecasts up to 28 days. In parallel, the 1/12° prediction system will benefit from an explicit scale separation in the data assimilation scheme to better exploit the observational information available for all scales. A global AI based forecast system (El Aouni et al, 2024) will also be put in operation. In the long run (post 2028), a higher resolution global system at 1/36° with improved numerical schemes and parametrizations should be put in operation. One of the priorities will be to introduce new wave-ocean and atmospheric boundary layer (ABL) couplings. The Northeast Atlantic model will include zooms at 1/108° to better represent dynamics in coastal areas.

The evolution of ocean prediction systems depends on the implementation of an adequate ocean observing system. While satellite altimetry has played and will continue to play a prominent role in ocean prediction, improving the altimeter constellation as discussed in section 4 is obviously not sufficient. It is imperative to enhance the integrated satellite and in-situ observing system as a whole to ensure that ocean prediction capabilities evolve to best meet societal needs.

*Author contributions.* PYLT led the paper, wrote and structured the manuscript. GD wrote section 2.1. MIP wrote section 2.2. JML contributed to section 2.4 and prepared Table 2. All other authors provided comments and corrections to an initial version of the paper.

*Competing interests.* The authors declare no competing interests.

*Acknowledgements.* The Copernicus Marine Service is implemented by Mercator Ocean International through a contributing agreement with the European Commission. The long-term support of CNES to the development of the DUACS system and its links with Mercator Ocean International forecasting systems is greatly acknowledged.

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
