# Peer review of "Satellite altimetry and operational oceanography: from Jason-1 to SWOT"

_EGUsphere, 2025_

## Author Comment (AC3)

**Temporal evolution of the RMS of the altimetry data (SLA). The figure shows the same seasonal cycle as observed in the innovation metrics presented in the paper.**

---

## Author Response (AR1)

**Reviewer 1**

**We thank the reviewer for his/her constructive comments. We have tried to take all of them into account in the revised version of the manuscript.**

This manuscript provides a valuable review of satellite altimetry, highlighting the value of the multi-satellite constellation of altimeters both for developing gridded products and as constraints for altimetry. The manuscript offers useful information about the potential requirements for future altimeter systems. The manuscript will be of interest to readers.

Prior to publication a number of issues should be considered, addressing both content and style.

1. Paragraph beginning line 180 and line 259. Mapping algorithms are designed to map anomalies relative to a zero mean, so there's an obvious challenge in working with data containing a 30-year trend. The text should talk about strategies and possible challenges in mapping data over an extended time period. The text specifically discusses using an altimeter reference period from 1993-2012. The manuscript should identify the potential challenges in relying on that reference period to map data in 2025, given the persistent sea level rise over this study period.

Mapping algorithms indeed assume that data have a zero mean. In the DUACS L4 processing system, the mean of the data selected for a given grid point estimation is first removed and then added back to the optimal interpolation result. We have added the following statement in the revised version of the ms "Note that, as mapping methods are designed to map anomalies relative to a zero mean, data selected for a given grid point estimation are recentered before the L4 mapping. Mean bias is then reinjected in the solution. The integrity of large-scale spatial and temporal signals (e.g., ocean level trends) is thus ensured."

2. line 55. "Only altimetry can constrain the 4D mesoscale circulation in ocean models which is required for most operational oceanography applications." This is a strong statement and could use a citation. Readers might reasonably expect that Argo can constrain 4D circulation, and that sea surface temperature or ocean color can constrain mesoscale circulation.

Argo does not resolve the mesoscale circulation but is essential to constrain the large scale 4D circulation. SST (and Ocean Colour) data do resolve the mesoscale circulation (depending on cloud cover except for SST microwave sensors) but SST represents surface dynamics and cannot be used alone to infer deep signals. Our statement was indeed ambiguous. Only the combination of altimetry with sparse in situ data and satellite sea surface temperature data can resolve the 4D circulation. The statement was rephrased as follows "Satellite altimetry plays a fundamental role to constrain the 4D...".

3. line 138, line 426, line 429, and elsewhere in the text. "wide-swath". SWOT's 120-km wide swath is not really wide in comparison to scatterometer or sea surface temperature swaths. Thus this terminology (although widely used) seems like a misnomer that will potentially leave readers confused and uncertain about the trade-offs between different altimeter options. Admittedly SWOT's swath is wider than ICESat-2's 9-beam laser system. Nonetheless, I'd suggest using "swath" rather than "wide-swath" or "wide swath".

Noted. We have used swath altimetry in the paper.

4. line 295. "SEEK (Singular Evolutive Extended Kalman filter) analysis". Given the central role that the SEEK algorithm plays, the manuscript should provide a citation when it introduces SEEK so that readers have a source to trace the foundations of the method.

We have added the following reference : Brasseur, P. and Verron J.: The SEEK filter method for data assimilation in oceanography: a synthesis, J. Ocean Dynamics, 56, 650– 661, https://doi.org/10.1007/s10236-006-0080-3, 2006.

5. line 305. In discussing the Mercator Ocean system, the manuscript should also discuss the forecast window. How long are model runs? Are there discontinuities between different runs, or are they smoothed to remove discontinuities?

We have added the following paragraph "The observations are assimilated with 7-day assimilation cycle and the forecasting systems deliver daily 10-day forecasts. All the increments coming from the analysis are applied progressively using the incremental analysis update (IAU) method (Benkiran and Greiner, 2008) which makes it possible to avoid model shock every week due to the imbalance between the analysis increments and the model physics, and results in an optimal and continuous model trajectory."

 More explanation is given in Lellouche et al. (2023) and illustrated on Fig. 4. of this paper (https://os.copernicus.org/articles/9/57/2013/os-9-57-2013.pdf): "Following the analysis performed at the end of the forecast (or background) model trajectory (referred to as "FORECAST" first trajectory, with analysis time at the $4^{th}$ day of the cycle), the IAU scheme rewinds the model and starts again from the beginning of the assimilation cycle, integrating the model for 7 days (referred to as "BEST" second trajectory) with a tendency term added in the model prognostic equations for temperature, salinity, sea surface height, horizontal velocities and sea-ice. The tendency term (which is equal to the increment divided by the length of the cycle) is modulated by an increment distribution function shown in Fig. 4."

6. line 392. "significant improvement". How significant is the improvement achieved by switching from 1 Hz to 5 Hz? A graphic or a quantification through rms differences is needed.

The use of 5 Hz data together with a refined along-track filtering of SLA reduced the analysis and forecast error variance by a few %. This is now precised in the text.

7. line 394. "seven altimeters". The results shown earlier in the manuscript explore the benefits of switching from 3 to 4 nadir altimeters. What are the relative gains in going to 5, 6, or 7 nadir altimeters? Is there a point of diminishing returns? Are there accuracy requirements that emerge when additional altimeters are available?

Adding more conventional altimeters improves the quality of analyses and forecasts. The effect is larger with high resolution models (e.g. moving from 1/4° to 1/12°). The impact indeed diminishes when adding more altimeters. The main issue is that one needs to improve both the spatial and temporal sampling. When trying to constrain the small space and time scales one also needs to consider that signals are much smaller (red spectra) and the altimeter noise becomes more important. In the 1/12° global model, there is always a small gain when adding more altimeters although the effect decreases as more altimeters are added. Analysis error variances (estimated by comparison with a non-assimilated nadir altimeter) are thus typically reduced by 10-15% when moving from 3 to 6 altimeters.

We have added the following sentence in the revised ms ""Adding more altimeters has always a positive effect on system performance, although the effect decreases as more altimeters are added. For example, analysis error variances (estimated by comparison with a non-assimilated nadir altimeter) are typically reduced by 10-15% when moving from 3 to 6 altimeters."

8. line 420. "The objective proposed for the Copernicus Marine Service (MOi, 2024) would be to resolve wavelengths larger than 50 km every 5 days." At what point is it worth considering swath measurements of surface currents (e.g. ODYSEA) which are possible over a scatterometer-scale swath, allowing more frequent revisits from a single satellite? It would be useful to be clear about the tradeoffs between total surface current and sea surface height.

The monitoring of total surface currents from space offers many exciting and challenging perspectives. TSCV observations cannot replace SSH observations but are quite complementary. The role of altimetry (conventional and swath) is to constrain balanced motions which is already quite demanding in terms of space/time sampling. TSCV includes balanced motions and non-balanced motions which have very different characteristics (time scales, vertical scales). The combination of TSCV and SSH observations will allow the estimation of both balanced and non-balanced signals which is required to best assimilate TSCV observations in a model.

9. line 435-436. improvement "by 15 to 20%". How much improvement would be obtained by instead going from three nadir altimeters to four nadir altimeters, i.e. the case discussed earlier in the manuscript?

This is less. Adding a fourth altimeter would reduce error by less than 10%.

10. lines 468-417. The final paragraph offers an overarching goal to emphasize the importance of the satellite and in situ observing system. It is surprisingly nonspecific, given the earlier recommendations for at least four altimeters and for swath capabilities. Perhaps the authors intend recommendations for multiple satellites to be encompassed by the words "consolidation and improvement". Or maybe the authors are trying to say the in situ observing system also needs attention. Whatever the goals, the authors should spell them out more explicitly in their conclusions by providing concrete recommendations.

We focus here on satellite altimetry. The main message is that we need to advance in parallel the other components of the satellite observing systems (eg SST, TSC) and of course in situ observing systems. Improving altimetry only is obviously not sufficient. We revised as follows the final paragraph:

« The evolution of ocean prediction systems depends on the implementation of an adequate ocean observing system. While satellite altimetry has played and will continue to play a prominent role in ocean prediction, improving the altimeter constellation as discussed in section 4 is obviously not sufficient. It is imperative to enhance the integrated satellite and in-situ observing system as a whole to ensure that ocean prediction capabilities evolve to best meet societal needs. »

11. Questions of grammar or style.

line 16. "have regularly improved". Wording is ambiguous can be clarified. For example, "satellite altimetry capabilities have improved through refinements in geophysical corrections, ...."

Rephrased as "have been regularly improved through refinements in geophysical corrections, processing algorithms including real-time processing, evolution of altimeter radar technology...

line 18. "thanks to the use of multiple altimeters and now swath altimetry." It's not clear if this phrasing applies to resolution or to everything about satellite altimetry. I'm guessing that only resolution is applicable. I would recommend starting a new sentence: "Resolution has also

improved through the use of multiple altimeters and now swath altimetry." (The use of "thanks to" is not wrong, but it is sort of a Gallicism.)

Corrected

line 19. "serving now" --> "now serving"

Corrected

line 22. Add comma after "system". The use of the so-called Oxford comma will help clarify that the DUACS system and the EU Copernicus Marine Service are not modifiers of Mercator.

Corrected

line 24. "in conclusion" --> "in the conclusion"

Corrected

line 28. "were flying" --> "flew"

Corrected

line 40. "was phased" --> "was phased in" or "began"

Corrected

line 40. "real time" --> "real-time"

Corrected

line 41. "It was also the start of Argo the global array of profiling floats that was required" --> "The start of Argo, the global array of profiling floats, was required"

Corrected

line 45. "high frequency" --> "high-frequency"

Corrected

lines 46 and 47. 2 instances: "a better" --> better"

Corrected

line 50. "consolidation and the": I would omit these words. It's not clear what the authors mean by "consolidation"

Corrected

line 51. "to constrain the" --> "in constraining". But really altimetry does not constrain the ocean. Rather it constrains the assimilation. So this should say, "Satellite altimetry plays a prominent role in constraining data assimilation models that represent the 4D ocean circulation."

Corrected as "Satellite altimetry plays a prominent role in constraining ocean models through data assimilation »

line 71. "builds and extends" --> "extends". Adding "builds" to this seems redundant.

Corrected

line 73. Add comma after "system"

Corrected

lines 81-82. "oceanography: geodetic". The structure used here with a colon is not clear. Change to "not optimized for oceanography because they employ geodetic missions, have nonoptimal payloads for orbit determination, or lack a microwave radiometer or dual frequency system, for example." In formal writing, dots (ellipsis) are used only for omissions from quotes, so are not appropriate here.

Corrected

lines 90-91. "In other words, ten years ago, the altimeter constellation was in a very fragile state, and operational oceanography was using a majority of non-operational altimeters." This sentence feels redundant and can be merged with the sentence starting "By 2012".

Paragraph revised as follows « In that context, there has been massive changes in the radar altimeter constellation over the past 15 years, both in terms of number of satellites and technology. From 2010 to 2015, the altimeter constellation was in a very fragile state, and operational oceanography was using a majority of non-operational altimeters. In 2012, altimeters of the first couple of modern generations had been decommissioned and only Jason-2 remained. Thanks to the coordinated efforts of space agencies and operational oceanography, the critical number of 3-4 satellites was secured with... »

line 90. "ten years". Is it 10 years or 13 years? The sentence starting at line 86 implies that the changes in the system started more than 10 years ago. This should be clarified.

Corrected (see above)

line 91. "outage" --> "outages"

Corrected

line 92. "a decent" --> "decent"

Corrected

lines 108-109. two instances: "one third is" --> "one third are"

Corrected

line 114. "get" --> "obtain". The word "get" is too casual for formal writing.

Corrected

line 129. "or resolution" --> "and better resolution"

Corrected

line 133. "bathymetries" --> "bathymetry"

We kept the plural (several new bathymetry products)

line 138. "tides model" --> "tide models"

Corrected

line 161. "reprocessing" --> "rounds of reprocessing" or "iterations of reprocessing"

Corrected

line 167.  "Faugere". In the author list, Faugere has an accent.  Should there be an accent in the references as well?

Corrected

line 184.  "at L3 level" --> "at the L3 level"

Corrected

line 205.  "at regional" --> "at the regional"

Corrected

line 253.  "as better" --> "as a better"

Corrected

line 266.  "get" --> "obtain"

Corrected

line 269.  "serving now" --> "that now serve"

Corrected

line 270.  Omit "now", since the word appears in the previous sentence.

Corrected

line 312.  "innovations".  This seems to be the first introduction of this technical term, so it should be defined here.  (Innovation is defined at the next usage.)

we removed here "integrate innovations".

line 350.  "require to have"  --> "require"

Corrected

line 396.  "quite".  I'd omit "quite".  The word can be either "very" or "not very".  This level of ambiguity is not helpful for readers.

Corrected

line 406.  "with a revisit time of 10 to 20 days".  This phrase is out of place in the sentence and appears to refer to wavelengths below 150-200 km.  Instead move it earlier in the sentence:  "A constellation of four altimeters with a revisit time of 10 to 20 days cannot resolve wavelengths below 150-200 km."

Rephrased as "A constellation of 4 altimeters cannot resolve wavelengths below 150-200 km and time scales of 10 to 20 days. «

line 408.  "Small scale" --> "Small-scale"

Corrected

line 433.  "performances" --> "performance"

Corrected

line 435.  "3 nadirs"  --> "Compared to results from three nadir altimeters, analyses and forecasts that add SWOT improve by 15 to 20%."

Corrected

line 438 and line 463.  "On the longer run"  --> "In the long run".  ("On the run" implies fleeing from the authorities.)

Corrected

line 443.  "consolidated".  What is meant by consolidated?  The term applies some strategic plan to optimize the constellation, but I don't have the sense that there is particular orbit planning underway.  Maybe it would be more accurate to say "expanded"?

Corrected (expanded)

line 457.  "couplings"  --> "coupling"

Corrected

lines 458 and 466.  "zooms"  --> "regional domain", "nested domain", "regional subdomain".  (In English, the word "zoom" doesn't really make sense as a noun to refer to a smaller domain.)

Corrected

line 459.  "AGRIF".  This appears to be the first and only introduction of an acronym.  What does it mean?

AGRIF (Adaptive Grid Refinement In Fortran) is a library that allows the seamless space and time refinement over rectangular regions in NEMO.

lines 469-470. "consolidation and improvement".  I think "consolidation" can be omitted.

Corrected

**Reviewer 2**

Review of the Manuscript: "Satellite altimetry and operational oceanography: from Jason-1 to SWOT"

**We thank the reviewer for his/her constructive comments.  We have tried to take all of them into account in the revised version of the manuscript.**

General Comments:

This manuscript provides a review of the evolution of satellite altimetry and operational oceanography, covering advancements from Jason-1 to SWOT. The historical perspective is valuable, particularly in demonstrating technological progress and its influence on ocean prediction systems. However, the reference list is dominated by a limited group of authors. A broader literature review, incorporating more original research rather than previous reviews by the same authors, would strengthen the manuscript.

The manuscript is well-structured and presents valuable insights into the evolution of altimetry and ocean modeling. However, addressing citation bias, refining certain phrasings, and incorporating more quantitative assessments will enhance its scientific impact.

Several references from the wider international efforts have been added in the introduction part. More quantitative assessments have been given (see also answer to the other reviewers) (altimeter noise SAR versus LRM, impact of multiple altimeters, impact of 5Hz versus 20 Hz)

Specific Comments:

- Lines 12 and 14: The term "development" appears twice; consider rewording for variation.

  Rephrased as "The development and evolution of satellite altimetry and operational oceanography are very closely linked. By providing all weather, global and real time observations of sea level, a key variable to constrain ocean analysis and forecasting systems, satellite altimetry has had a profound influence on the advancement of operational oceanography. »

- Line 19: Argo should be mentioned as a key observing system.

  Argo is mentioned in the introduction.

- Line 24: The word "discussed" is repeated; rephrase for clarity.

  corrected.

- Line 31: Clarify that Jason-1 was a reference mission (following Topex/Poseidon), critical for long-term sea level studies and cross-calibration with other missions.

  Rephrased as "The launch of Jason-1 in 2001 was then a major milestone to start a series of highly precise long-term reference missions (following T/P) with near real time processing capabilities. »

- Line 43: Rephrase "Many improvements have been made" and include additional references on operational oceanography and modeling.

  Rephrased as "There have been major advancements in operational oceanography systems"

- Lines 51 and 54: The phrases "prominent role to constrain the 4D ocean circulation" and "strong constrain for inferring the 4D ocean circulation" are repetitive; consider rewording.

  Done. See also answer to reviewer 1.

- Line 61: The word "dramatically" may be too strong; consider a more precise alternative.

  Replaced by greatly improved.

- Line 72: Cite original research rather than relying solely on review papers.

  We do not see how this can be done here in the introduction section. Original research papers are cited in the following sections.

- Line 85: Replace "massive." Additionally, should this statement reference the last 20 years rather than 10? The following sentence begins with 2012, suggesting a broader timeframe.

Replaced by considerable. Paragraph was also revised following reviewer 1 comments (e.g. 15 years instead of 10 years).

- Line 88: The critical number of 3-4 satellites—include a supporting reference.

Le Traon et al. (2017).

- Line 130: Again, reconsider using "massive."

Major.

- Line 166: Why start from 2010? The review covers the last 20 years.

This is an update wrt the previous review in Dibarboure et al (2011).

- Line 173: The word "pertains" is awkward; consider rephrasing.

Another area of improvement is the refinement...

- Line 178: Including a figure illustrating correlation scales of different products would be valuable. If a figure is not feasible, a paragraph discussing resolved scale values should be added.

This paragraph has been completed as follow, integrating also comments from reviewer 1: "In the current version of the L4 product (i.e. DT-2024), the average spatial resolution of the global gridded SLA is around 180 km at mid-latitudes, with larger values near the equator (~500 km) and finer values at the poles (~100 km). Compared with the previous DT-2014 version (discussed in Ballarotta et al, 2019), the effective spatial resolution has been improved by almost 10% at mid-latitudes; the bulk of the improvements being made by changes in the DT-2024 version with a significant improvement induced by the MIOST mapping method. Note that, as mapping methods are designed to map anomalies relative to a zero mean, data selected for a given grid point estimation are recentered before the L4 mapping. Mean bias is then reinjected in the solution. The integrity of large-scale spatial and temporal signals (e.g., ocean level trends) is thus ensured.

- Line 192: The section title "Assimilation" is unnecessary; consider removing it.

Section titles reprocessing for mesoscale, climate and assimilation) have been removed

Line 195: "Product content has evolved over the years"—reword for clarity.

L3 sea level products used for data assimilation have evolved over the years.

- Line 202: "SAR measurements... greatly reduce noise"—quantify the reduction. Several validation studies using in-situ observations (e.g., tide gauges, drifters, gliders) should be referenced as well.

This sentence was revised as follows "Since November 2023, a major milestone in the evolution of DUACS products has also been reached, with an increase in product resolution. The NRT sea level system now takes the 20 Hz initial resolution data as input, rather than the 1 Hz resolution previously used. This was made possible by the advances in altimeter techniques and processing, such as the Synthetic Aperture Radar (SAR) measurement mode, which allows for a significant reduction in measurement noise (e.g., by about 30% as underlined by Raynal et al., 2018). Additionally, specific empirical noise

mitigation corrections enable the reduction of measurement noise in conventional LRM measurements (Tran et al., 2021)."

- Line 211: Add references to support model forecasts.

  We removed the sentence "Because the assimilation of sea level from altimetry data remains essential for the accuracy of model forecasts »

- Section 2.4: Expand the discussion to include additional operational oceanography systems, not only from Europe.

  Several references added

- Suggested Addition on DTO to the discussion (with help from ChatGpt, so please revise):

  We have explained the role of AI for ocean prediction and added a sentence of the role of DTOs for operational oceanography.

  "AI is a rapidly evolving field and has a major potential for ocean and sea ice forecasting and to improve analysis and forecasting systems (e.g. model emulation, subgrid scale parametrisation, calibration and bias correction) (e.g. Heimbach et al., 2024).

  By providing dynamic and interactive cloud-based platforms to model and predict the ocean, explore various scenarios, assess impacts to make informed decisions, ocean digital twins are transformative technologies that will offer new perspectives for the development of operational oceanography."

As the field of operational oceanography progresses, the concept of a Digital Twin of the Ocean (DTO) offers a promising advancement in how ocean dynamics are monitored, modeled, and managed. By integrating real-time satellite altimetry data from missions such as SWOT with other ocean observations, a DTO provides a detailed, dynamic representation of the ocean's state. This virtual system combines data from multiple sources—including altimetry, ocean circulation models, and in-situ measurements—to offer a more integrated and predictive approach to ocean monitoring. The role of in-situ observations is particularly crucial, as they provide direct, high-accuracy measurements that complement satellite data, ensuring the DTO's simulations are grounded in real-world conditions.

The DTO complements the operational capabilities of SWOT by not only providing a snapshot of current ocean conditions but also simulating future ocean states. Through advanced modeling, a DTO can predict changes in ocean dynamics, such as sea surface height, currents, and temperature, based on real-time observations from SWOT, in-situ data, and other sources. The use of AI and machine learning models further enhances these predictions by improving the assimilation of data and refining the accuracy of simulations. This would significantly enhance the operational use of satellite altimetry data for ocean forecasting and management.

Furthermore, the DTO could serve as a key decision-support tool for stakeholders in ocean management, climate adaptation, and disaster response. By linking SWOT's high-resolution data with predictive ocean models, a DTO would enable more accurate forecasting of oceanic conditions, improving the management of coastal areas, fisheries, and marine resources. Additionally, the integration of DTOs could provide feedback on the calibration of satellite measurements, improving the accuracy of future observations.

Incorporating the Digital Twin of the Ocean into operational oceanography represents a crucial step forward, offering both enhanced predictive power and a more comprehensive understanding of the ocean's role in the broader Earth system.

Figures:

- Figure 1: Update the figure, as it is now outdated.

  Figure 1 updated

- Figure 2a: The image is unclear; consider enlarging it for readability.

  Figure updated (quality) (higher resolution)

- Figure 2b: This figure is relevant—consider extending it back to 2001 (Jason-1 launch: December 2001) to provide a more complete historical perspective, aligning with the title "Satellite altimetry and operational oceanography: from Jason-1 to SWOT."

  Figure 2b updated as recommended.

**Reviewer 3**

First of all this is a very nice and timely article reviewing the status of assimilation of altimetry data for operational oceanography. I only have a couple of technical question and a request and the rest are minor editorial comments.

**We thank the reviewer for his constructive comments.  We have tried to take all of them into account in the revised version of the manuscript.**

I am not an expert of data assimilation, but I am puzzled by the extremely small magnitude of innovation, only a few mm, as shown in Fig 6.  In the context of Kalman Filter, the innovation indicates the extent of the data affecting the model trajectory.  Such a small innovation relative to the measurement errors seem to indicate that the model estimates are not much affected by the data, also reflected in the sizable model misfit.  I may be wrong, but this seems to me that either the assimilation time step is too small, or the error covariances are not properly specified.  In any case, some elaboration of the magnitudes of the innovation and model misfit is warranted.  Are the results satisfactory? Why?

Rms of the innovations (model forecast minus observations) are of about 5 to 6 cm.  The correction (i.e. increments calculated from the innovations) applied to the model is thus quite significant.  The mean innovation is a mean over the global ocean. The figure just allows us to check that we do not have a mean bias between observations and model forecasts.

In Fig 6, there is an apparent annual cycle in both the innovation and misfit time series. Explanations?

The seasonal cycle in the innovation is also observed in the SLA variability. It is most likely related to the evolution of the altimeter coverage due to the ice cover in the Arctic. This is now explained in the revised version of the paper.

In the supplementary material, a figure shows the temporal evolution of the RMS of the altimetry data (SLA), showing the same seasonal cycle observed in the metrics presented in the paper. We

don't want to add this figure to the paper but just wanted to support our answer with an illustration.

[Figure]

The paper is pretty much Euro-centric,  I think it is worth mentioning other significant efforts such as ECCO of the US.

The paper is indeed euro centric and this is explicitly stated in the introduction.  However, several references (incl. Heimbach et al., 2019)  from the wider international efforts have been added in the introduction part.

Heimbach, P., Fukumori, I., Hill, C. N., Ponte, R. M., Stammer, D., Wunsch, C., et al. (2019). Putting It All Together: Adding Value to the Global Ocean and Climate Observing Systems With Complete Self-Consistent Ocean State and Parameter Estimates. *Frontiers in Marine Science*, *6*, 55. https://doi.org/10.3389/fmars.2019.00055

Minor comments:

Line 28: change 2011 to 2001

 Done

Line 41: change "mandatory" to "essential"

Corrected

Line 45: provide references for the claim "Regional systems have reached a resolution of a few kilometers"

References added

Lines 44-49: need some references

 References added

Lines 50-63:  This information belongs to the beginning of the introduction.

The first 2 sections are focused on altimetry and operational oceanography and the third section on the links between altimetry ad operational oceanography.

Line 121: change "the explore" to "explore"

Corrected

Lines 119-125:  Some references are needed for the science advance from SARM.

The reference to Raynal et al 2018 has been added.

Table 1: What are the meanings of the information of second and third rows?

This is now explained in the legend. « The mean timeliness for Copernicus 1 phase 1 (2015-2017), Copernicus 1 phase 2 (2018-2021) and Copernicus 1 (2015-2021) are also given »

Lines 248-249: need references on GOCE and GRACE

The following reference was added : Flechtner, F., Sneeuw, N., Schuh, WD. (eds) Observation of the System Earth from Space - CHAMP, GRACE, GOCE and future missions. Advanced Technologies in Earth Sciences. Springer, Berlin, Heidelberg. https://doi.org/10.1007/978-3-642-32135-1, 2014.

Line 251: change "hydrological" to "hydrographical"

Replaced by temperature and salinity

Line 295:  Need references on SEEK

Added.

Lines 335-340: Please provide the top applications of the system and some discussions.

We have added a link to Copernicus Marine use cases : https://marine.copernicus.eu/services/use-cases

Fig 6:  what is SWON?

SWOT nadir.  This is not explained in the legend of the figure.

Line 459:  Please explain what AGRIF represents.

Adaptive Grid Refinement In Fortran.  This is now explained in the text.

---

## Author Response (AR2)

**All final minor corrections required by reviewer 1 have been taken into account in the final version of the paper**